# Food Fraud Vulnerability Assessment in the Chinese Baijiu Supply Chain

**DOI:** 10.3390/foods12030516

**Published:** 2023-01-23

**Authors:** Yiqin Wang, Jiali Liu, Yanling Xiong, Xuefan Liu, Xiaowei Wen

**Affiliations:** 1College of Economics and Management, South China Agricultural University, Wushan Road 483, Tianhe District, Guangzhou 510642, China; 2Research Center for Green Development of Agriculture, South China Agricultural University, Wushan Road 483, Tianhe District, Guangzhou 510642, China

**Keywords:** food fraud vulnerability assessment, baijiu supply chain, opportunities, motivations, control measures

## Abstract

As a representative of Chinese alcoholic drinks, baijiu has developed into a mass-consumption commodity. Its simple industrial chain makes it a suitable target for fraudsters. In order to understand the differences and potential factors of fraud vulnerability among groups at various levels, this study constructed a food fraud vulnerability assessment system for the Chinese baijiu supply chain based on routine activities theory. We examined the fraud vulnerability in the baijiu supply chain with data from 243 producers and 45 retailers by using the safe supply of affordable food everywhere (SSAFE) food fraud vulnerability assessment (FFVA) tool. The results indicate that fraud factors related to opportunities have an overall medium vulnerability, while those related to motivations and control measures have an overall medium-low vulnerability. In addition, there are significant differences in the perceived vulnerability of fraud factors across the supply chain. Moreover, retailers have overall higher fraud vulnerability in terms of opportunities and control measures than producers. The main reasons for the frequent occurrence of fraud in the baijiu industry are numerous technical opportunities, strong economic drivers, and insufficient control measures.

## 1. Introduction

As the world’s largest developing country and in a special stage of economic and social transformation, China’s current food safety situation is more severe, and the proportion and impact of food fraud incidents are also increasing [1,2]. Food fraud has existed since ancient times. In ancient Rome and Athens, there were cases of adulteration of wines by adding sweeteners and coloring agents [3,4]. The Global Food Safety Initiative, the Grocery Manufacturers Association in the United States, the US Pharmacopia, and reports of the European Parliament all regard food fraud as a deliberate and deceptive action, committing fraud for financial gain [5]. The former Food and Drug Administration of China issued *the Measures for Investigating Food Safety Fraud*, which defines food safety fraud as actions that businesses or individuals intentionally provide false information or hide the truth in any sector of multiple supply chains, such as production, storage, transportation, sales, and food service. Academics generally consider food fraud to be a deliberate deceit that sellers use to obtain higher economic benefits through illegal acts; for instance, unclear labeling, misrepresentation of ingredients, and unlawful additions [3]. Therefore, this study defines food fraud as a series of dishonest behaviors including intentional adulteration, false propaganda, substitution, or other behaviors made intentionally by individuals or organizations in the food market for their own benefit.

With the expansion of world trade, emerging markets spring up, and global food prices steadily rise [6]. The types of food fraud have become increasingly complex. People’s safety of life and property are under increasing threat from food fraud, and it also poses a serious threat to the profits of agri-food companies, producers, and processors [7,8,9]. The Grocery Manufacturers Association estimated that global economic loss associated with food fraud is between $10 billion and $15 billion each year [10]. In 2017, joint coordinated operations by the International Criminal Police Organization (INTERPOL) and Europol resulted in the confiscation of approximately 10,000 tons of counterfeit food products worth EUR 230 million in a total of 61 countries [11].

Understanding the vulnerability of food fraud and its extent and assessing the factors that lead to food fraud are necessary steps to prevent and mitigate food fraud or adulteration [12]. In the field of food fraud research, fraud vulnerability refers to the existence of weaknesses in the system that can provide opportunities for fraudsters. It also represents the risk grade of food fraud in the food supply chain. Consumers might be exposed to health risks if no measures are taken to control them [13]. Thus producers, processors, and retailers are becoming concerned about food fraud detection and prevention, and the social communities are beginning to focus on developing food fraud vulnerability assessment methods [3]. In view of the situation, a specialized food fraud vulnerability assessment (FFVA) tool has been created by Safe Supply of Affordable Food Everywhere (SSAFE) in collaboration with PricewaterhouseCoopers, Wageningen University, the Free University of Amsterdam, and other institutions based on the routine activities theory. The tool is used to identify the areas where food fraud is likely to occur in the supply chain and clarify the strengths and weaknesses of strategies which could mitigate the current situation of food fraud in the organization [10,14]. The FFVA is divided into two dimensions with a total of 50 questions. One dimension includes three factors that influence criminal behavior: opportunity, motivation, and control measures. The other dimension addresses the internal and external environmental factors of the firm, including direct suppliers and consumers, the broader supply chain or industry network, and the domestic and international governance environment [9]. This assessment tool has been tested multiple times by the global food industry, government, scientific community, and other organizations [15]. It has successfully assessed the fraud vulnerability in the spice supply chain [16], the Dutch milk supply chain [17], and the extra virgin olive oil supply chain [18].

To effectively mitigate fraud in the marketplace needs to strengthen multi-entity regulation and enforcement [19]. Spink et al. summarized public policies for the prevention and management of food fraud through an assessment of the steps in the development of policies related to food fraud [20]. By constructing an evolutionary game model, Wang et al. found that firms are less willing to engage in adulteration when they have long-term interest goals [21]. Cao et al. constructed an asymmetric evolutionary game model between the government and food companies and found that the government should strengthen the behavioral regulation of new media in effectively controlling adulteration in the food market [22]. 

Scholars in various countries have relatively abundant research on food fraud, food fraud vulnerability, vulnerability assessment, and food fraud problem governance. However, most of them focus on a single level, such as definition, connotation, types, characteristics, impact, assessment, and solution countermeasures. Research on food fraud in China has been fragmented, focusing mostly on the sorting out of food fraud research and the simulation of food fraud research subjects. To the authors’ knowledge, there are relatively few studies that have been conducted specifically on a particular type of food from a food fraud vulnerability perspective. Therefore, this study conducts a food fraud vulnerability assessment of the baijiu supply chain.

Liquid foods are more prone to fraud than solid foods [23], so alcoholic beverages are one of the objects vulnerable to fraud [24]. Chinese baijiu is currently the world’s best-selling liquor, with sales exceeding 10 billion liters in 2018 [25]. According to statistics, alcohol fraud has ranked second among food fraud in China in the last 20 years [26]. Incidents of baijiu fraud are frequently reported [27]. However, it is unclear which factors contribute to the vulnerability of the baijiu supply chain and whether there are differences in fraud vulnerability among participants. Therefore, this study uses the SSAFE FFVA tool to evaluate and analyze the vulnerability of baijiu food fraud factors. The analysis would help us explain the mechanism of food fraud and find the deep-seated reasons for the outbreak of food fraud. To strengthen the governance of food fraud issues and better promote the quality development of the food industry, it might well put forward some countermeasures and recommendations.

## 2. Theoretical Analysis

This study sorts out the mechanism of food fraud generation based on the routine activities theory in criminal behavior (Figure 1). The theory suggests that three conditions need to be satisfied for criminal activity to arise: (1) suitable targets; (2) motivated offenders; and (3) the absence of capable guardians. The food market satisfies exactly these three conditions, i.e., suitable targets (profitable food), criminally motivated offenders (unscrupulous businessmen willing to make profits through fraudulent activities), and lack of competent monitors (inadequate regulation and penalties such as the food industry or relevant law enforcement agencies, etc.). The FFVA tool considers food fraud as a potentially criminal act, and the three key factors influencing the occurrence of food fraud are: opportunities (suitable target), motivations (motive to commit the crime), and control measures (lack of prevention and regulatory-related measures). The underlying theoretical framework is as follows.

## 3. Materials and Methods

### 3.1. The Food Fraud Vulnerability Assessment (FFVA) Tool

The SSAFE FFVA tool is commonly used for questionnaire design. Aside from the routine activities theory, “design rules” used in the evaluation and diagnosis tool of the food safety management system are also fused into the tool [28]. The “design rules” focuses on the key factors that lead to food fraud incidents and subdivide the key factors into a number of indicators for differentiated assessment [16]. To ensure that the FFVA tool could be applied to the actual situation of the Chinese baijiu supply chain, the original questionnaire was modified by the Delphi expert consultation method. The final food fraud vulnerability assessment system applied to the baijiu supply chain is shown in Table 1. Opportunities, motivations, and control measures in the FFVA tool can be further subdivided into six categories: technical opportunities (Factors 1–5), opportunities in time and space (Factors 6–9), economic drivers (Factors 10–13, 17, 18, 23, 26, and 27), cultural and behavioral drivers (Factors 14–16, 19–22, 24, and 25), technical measures (Factors 28–30, 35, 36, and 40), and managerial measures (Factors 31–34, and 37–39) [15].

### 3.2. Data Collection

According to a previous study, food fraud in the baijiu supply chain occurs mainly in the production and retail sectors, and there are a large number of questions existing in the upstream suppliers of each sector. Therefore, this study chose to investigate the baijiu supply chain and its tiers (producers and retailers). The baijiu producers involved in this study contain well-known brands of Chinese baijiu and small workshops of ordinary baijiu production. The retailers include large supermarkets such as Suning Tesco, designated direct baijiu stores, baijiu specialty stores, and small supermarkets. Respondents were from Guizhou, Shandong, Hubei, Anhui, Henan, Jilin, Guangdong, and other provinces with large baijiu production. Both field research and online research were used. A forward–backward translation process was employed to ensure the content validity of the scales’ translation to Chinese [29]. In addition, we invited relevant experts to pilot the questionnaire and provide feedback. Minor adjustments were applied based on their input. During the formal research, the researcher explained to the interviewee that the data obtained from the research were strictly confidential and would only be used for scientific research. During the research, the researcher asked the interviewee questions and explained the questions if necessary. Because the FFVA tool involves professional questions related to multiple corporate sectors, such as procurement, quality management, and human resource management. To ensure the accuracy and completeness of the questionnaire answers, field research required members of multiple departments to respond synergically. Our survey was also conducted on a popular professional online survey platform “Questionnaire Star” in China [30]. Questionnaires were distributed to middle and senior managers of baijiu enterprises in several provinces of China through the heads of food industry associations and related departments. Thus, it could guarantee a wide range of data sources for the questionnaires. The average time for field and online research is 1 h.

The survey obtained a total of 243 valid questionnaires from baijiu producers and 45 valid questionnaires from baijiu retailers. In the production sector, middle and senior managers account for 70.8%, employees in the production departments account for 15.2%, and employees in other departments account for 14.0% (see Table 2). Since the research in the retail sector is all field research, the interviewees, through targeted screening, are all middle and senior managers who have been in the baijiu retail industry for at least 5 years. They have a clear knowledge of the baijiu industry, which ensured the scientific rationality of the questionnaire data. Baijiu has a special production mode. Producers directly complete the production from raw materials to the final product. Therefore, the self-production and self-sales model of the baijiu workshops basically covers the retail sector of the supply chain. This results in a relatively small sample of pure baijiu retail stores. However, it was still possible to analyze the food fraud vulnerability of the entire baijiu supply chain accordingly.

### 3.3. Data Analysis

The data were analyzed in two aspects: first, the overall fraud vulnerability of the baijiu supply chain was analyzed by frequency and weighted frequency methods, and then the differences in fraud vulnerability between producers and retailers were analyzed using Multiple Correspondence Analysis (MCA) and Mann-Whitney U test.

#### 3.3.1. Frequency and Weighted Frequency Analysis

The data were analyzed using a three-level scale that classified most fraud factors into three levels: low, medium, and high. For the opportunity and motivation fraud factors, 1, 2, and 3 correspond to low, medium, and high levels of vulnerability, respectively. For the fraud factors of control measures, 1, 2, and 3 correspond to high, medium, and low vulnerability levels, respectively. In order to ensure that respondents fully understand the questionnaire, four options were set for some questions, but the final scoring was consistent with the above explanation. That is to say, the two ends of the scale were given one and three points, respectively, with two points assigned to the middle. According to the vulnerability levels of different factors, the overall vulnerability level of the baijiu supply chain could be obtained, and then the level of risk of food fraud in the baijiu supply chain would be judged.

In order to obtain the overall results of the baijiu supply chain, the study should ensure that the contribution of baijiu producers and retailers among the research interviewees was equal. Scores that meet the producer indicators are the final result with the highest scoring frequency among the producer indicators. The scores that meet the common indicators of producers and retailers are determined using the weighted frequency formula.
$$F_{i}=\frac{1}{2}\Sigma_{j}\frac{X_{ij}}{n_{j}}$$

$F_{i}$ represents the frequency of each indicator score i, $X_{ij}$ represents the number of observations of score i in group j (j = producers, retailers), $n_{j}$ represents the total number of observations in group j.

#### 3.3.2. Multiple Correspondence Analysis

MCA refers to the data dimensionality reduction of multiple associated variables (generally reduced to two-dimensional), and the corresponding relationship between variables is visually displayed in the two-dimensional space [31]. XLSTAT (version 2019.2.2, Addinsoft, Paris, French) is a powerful and flexible excel data analysis add-in that allows users to analyze, customize, and share results in Excel (2016, Microsoft, Redmond, Washington, USA), making it the tool of choice for statistical analysis in businesses and universities. The assessment indicators, applied to both baijiu producers and retailers, were conducted MCA by using XLSTAT 2019. Then, the study could obtain the center-of-mass coordinates of different assessment indicators for all research subjects. The results would be used to analyze the association of results between different groups in China’s baijiu supply chain.

#### 3.3.3. Mann-Whitney U Test

The Mann-Whitney U test is a nonparametric method of univariate analysis of variance (ANOVA) proposed by H.B. Mann and D.R. Whitney in 1947 to test whether there is a significant difference between the means of two aggregates. In contrast to parametric tests, nonparametric tests do not assume a normal distribution of the data and chi-squaredness, and do not have specific requirements for sample size. This study compared the difference in fraud vulnerability between producers and retailers by the Mann-Whitney U test (*p* < 0.05 was considered significant), which was partially conducted using SPSS (version 22.0, IBM, Chicago, IL, USA) software.

## 4. Results and Discussion

This section assesses the overall fraud vulnerability of the baijiu supply chain based on three factors: opportunities, motivations, and control measures. The key fraud factors would be identified in these factors. Meanwhile, the section also assesses the differences in fraud vulnerability factors between tier groups.

### 4.1. FFVA in the Baijiu Supply Chain: Overall Results

Only the common factors of both tier groups were used for making the radar chart, performing MCA and the statistical comparisons. The overall fraud vulnerability perceived by producers and retailers in the baijiu supply chain is shown in Figure 2 and Figure 3. The specific analysis and discussion are as follows.

#### 4.1.1. Opportunities

Food fraud is considered a criminal act; therefore, understanding why offenders commit food fraud is a prerequisite to tackling the problem of fraud in the baijiu industry [32]. As seen in Figure 2a, six of the nine opportunity fraud factors are rated as medium vulnerability (Factors 1–5, 8). Factors 6 and 7 are rated as low vulnerability. Factor 9 is rated as high vulnerability. Producers perceive most of the opportunity fraud factors as a medium vulnerability compared to retailers (Figure 3a). Overall, the opportunity fraud factors show medium vulnerability.

##### Technical Opportunities

Technical opportunities include the difficulty of adulteration in the process of baijiu production and packaging, the ability to detect counterfeiting of baijiu, etc. All fraud factors 1, 2, 3, 4, and 5 of the technical opportunity are medium vulnerability (Figure 2a). With the means of manufacturing fake wine improving, the technical threshold for counterfeiting raw materials and baijiu products is not high, but the detection means for counterfeit products are not mastered by everyone because there is no easy and inexpensive way to provide adequate identification and authentication [25]. Not only do the general public consumers have no detection capabilities, but many retailers who have been selling baijiu for many years are also unable to make accurate judgments about the authenticity of the product. For example, fake Maotai containing special chip packaging can pass supermarket screening and even professional inspection by the Maotai distillery, which makes it difficult for regulators to enforce the law and damages consumer confidence in the entire baijiu industry.

##### Opportunities in Time and Space

Opportunities in time and space refer to the possibility that the offender can act in dishonesty without being regulated. From Figure 2a, the control of the production and processing process and supply chain transparency are in low vulnerability (Factors 6 and 7). The study findings are parallel to the findings by Yang et al. [12]. The baijiu production process is strictly controlled, and personnel should register in and out of the majority of baijiu brewing, processing, and packaging areas, and production management personnel would be supervised. Baijiu producers and retailers mostly maintain longer-term business relationships with their upstream suppliers, while producers also have detailed knowledge of their downstream agents and distributors. Therefore, the control of the production and processing process and supply chain transparency are scored as low vulnerability. The historical events of raw material counterfeiting and final product counterfeiting are medium and high vulnerability, respectively (Factors 8 and 9). Counterfeit products are still in the market, and final product fraud occurs more frequently. The information is mostly publicly knowable. Therefore, the fraud element of opportunities in time and space shows a medium to a high level of vulnerability.

#### 4.1.2. Motivations

Motivation fraud refers to the forces and thoughts that trigger the offender to commit fraudulent acts. In addition, it also explains why the offender wants to commit fraud [32]. Of the 18 fraud factors related to motivations, 11 are rated as low vulnerability, 4 are rated as medium vulnerability, and 3 are rated as high vulnerability (Figure 2b). Both producers and retailers perceive most of the motivation-related fraud factors as low vulnerability (Figure 3b). Overall, the motivation-related fraud factors exhibit low to medium vulnerability.

##### Economic Drivers

First, as seen in Figure 2b, the financial pressure on suppliers and the economic situation of suppliers are low vulnerability (Factors 17 and 18). This result is consistent with most realities that the majority of producers and retailers are also customers of upstream suppliers in the baijiu supply chain. It causes little financial pressure on upstream suppliers, so most supplier firms could make a good profit. As a consequence, the risk of dishonest behavior by upstream suppliers in all sectors is low due to economic drivers.

Second, the raw material supply, the economic situation of the enterprise, the economic situation of the supply chain sector, and historical events of fraud in the supply chain are medium vulnerability (Factors 11, 13, 23, and 26). This may be because producers’ and retailers’ upstream suppliers have sufficient sources. In general, producers and retailers purchase large quantities of raw materials in advance for storage and trade with relatively regular suppliers. The retail sector is relatively well-stocked except for the Maotai series of products. Flying Maotai is vulnerable to suffering supply–demand imbalances in the retail sector, which often lacks supply and is unable to meet consumer demand, so Factor 11 is in medium vulnerability. Although the baijiu industry is in a mature stage with relatively stable development, the majority of producers and retailers believe that the current market is highly competitive and it is difficult for enterprises to obtain lucrative benefits in the long term. Low-end baijiu enterprises have a small market share, low profits, high economic pressure, and overall higher economic-driven risk in the baijiu supply chain. Therefore, Factors 13 and 23 are medium vulnerability. Although China has carried out corresponding special rectification actions against food fraud in the baijiu industry in recent years, there are many lengthy sectors in the baijiu supply chain, and fraud still occurs from time to time. Fortunately, most of the information is public, and thus Factor 26 is medium vulnerability.

Third, the raw material price, the specific composition or properties of raw materials, and the competition level in the supply chain are all high vulnerability (Factors 10, 12, and 27). As the most basic production material for baijiu, fluctuations in the price of raw materials mean changes from the source of the supply chain, which will inevitably lead to price changes throughout the chain. Finally, it will ultimately be reflected in the retail end (i.e., whether consumers buy it or not). An important driver of food fraud stems from economics since offenders commit food fraud by seeking to maximize profits or minimize losses; therefore, raw material prices are high vulnerability. The popular saying “grain is the meat of baijiu” in the baijiu market shows that the quality of grain can directly determine the quality of the original baijiu. For example, Guizhou Maotai spent 280 million to support the cultivation of Little Red organic sorghum. In addition, consumers are only able to assess the existence of certain attributes after consuming or using the product (empirical attributes). Sometimes consumers are unable to determine the existence of attributes even after consumption (trust attributes). These provide offenders with an incentive to commit fraud [33]. In recent years, there have been a number of fraud incidents in the retail sector involving operators counterfeiting Baijiu in “Luzhou” or other places in order to obtain higher interests. The essence of such a behavior is to claim that the quality of raw materials is high. In summary, for the baijiu business entities and consumers, the origin and other special attributes have a high value, so it is a high-risk point for the occurrence of food fraud.

##### Cultural and Behavioral Drivers

Cultural and behavioral drivers consist of three aspects, ethical culture, organizational strategy, and law-breaking situation. From Figure 2b, these three aspects of drivers of fraud are all low vulnerability (Factors 14–16, 19–22, 24, and 25). This is consistent with the vulnerability of the milk supply chain [12]. First, this result indicates that a good ethical, cultural environment can constrain and influence human behaviors. On the contrary, a bad cultural environment can easily lead to rampant illegal activities; therefore, maintaining an appropriate ethical business culture is conducive to preventing fraud, which is consistent with the results of existing studies [34]. Second, the whole supply chain should maintain a benign organizational strategy. Regardless of future changes in the environment, to ensure the production is healthy and sustainable in the long run, the production and operation subjects will naturally not risk committing fraudulent acts. Third, the whole supply chain should pay attention to the illegal and non-compliant issues that appear in each sector and actively take social responsibility, which may reduce the risk of the production and operation subjects committing fraudulent acts.

#### 4.1.3. Control Measures

Control measures are key factors in management activities, which mainly refer to the relevant actions that can detect or prevent the occurrence of food fraud. It is mainly used to mitigate and reduce vulnerabilities arising from fraud opportunities and motivations [15]. From Figure 2c, it shows that among the 13 control measure fraud factors, Factors 28 and 33 are rated as high vulnerability, Factor 39 is rated as medium vulnerability, and 10 control measure fraud factors are rated as low vulnerability (Factors 29–32, 34–38, and 40). Both producers and retailers perceived most of the control measure fraud factors as low vulnerability (Figure 3c).

##### Technical Measures

Technical measures refer to the measures that can prevent and mitigate food fraud, mainly by using technical means. Factor 28 shows high vulnerability, which is probably due to the fact that raw material counterfeiting detection systems are mainly used by producers, while most retailers do not test the authenticity of raw materials (Figure 2c). Factors 29, 30, 35, 36, and 40 are all low vulnerability. This is principally because there are different measures in several sectors of the supply chain to prevent food fraud. Before baijiu products enter the market, most producers will carry out random spot checks on the products to ensure that the quality is qualified, and the packaging is error-free. In the supplier sector, most enterprises have powerful traceability systems. Some enterprises that have not established traceability systems would collect and record detailed information from upstream direct suppliers to downstream customer enterprises. This method could also strengthen the control over sources and destinations of products. Without traceability in the food supply chain, experienced production companies will have the incentive to sell poor-quality products [35]. Direct suppliers upstream of the baijiu supply chain are set up with qualified food quality management processes, and some will test their products for counterfeiting, so suppliers upstream of each sector rarely experience food fraud. Once an enterprise encounters a consumer reflecting that a baijiu product is counterfeited, both producers and retailers will test the authenticity and the source of the product. Enterprises will formulate appropriate emergency response plans for different situations to ensure the interests of consumers while also maintaining their own products and enterprise image.

##### Managerial Measures

Managerial measures refer to organizational management behaviors such as ethical behavior constraints of the enterprise itself, upstream suppliers, and the food market. Factors 31, 32, 34, 37, and 38 are all low vulnerability (Figure 2c). The vast majority of enterprises will examine the ethical qualities of their employees when hiring them so as to eliminate the possibility of internal employees using illegal means to damage the quality of their products. Within each enterprise, a corporate ethical culture will be widely communicated to employees and a written code of ethics will be implemented to enhance the ethical literacy of internal employees, thereby preventing unethical behavior. Enterprises in each sector of the supply chain attach great importance to a moral and ethical culture. To maintain close cooperation with upstream suppliers, they sign mutual contracts that contain both product quality parameters and mutual integrity and moral constraints. Guidelines exist throughout the supply chain for the norms of behavior and prevention of food fraud incidents. Most enterprises have a high degree of self-regulation and there is a communication of information among them. Some enterprises also have associations that organize collective learning to prevent fraud. Factor 33 shows a high vulnerability. Some enterprises have set and implemented a good whistleblowing method. While increasing the protection of whistleblowers, it can increase the probability of detecting unethical behaviors such as internal counterfeiting. However, most enterprises have not set up corresponding internal whistleblowing systems, so they lack certain control safeguards over the occurrence of internal dishonesty. Factor 39 shows medium vulnerability. According to the interviewees, although there is local law enforcement against food fraud activities, the frequency of inspections by the relevant regulatory authorities is not high and the punishment is not harsh. Enforcement has not been implemented, especially in the townships. Law enforcement actions have yet to be further promoted.

### 4.2. FFVA in the Baijiu Supply Chain: Differences between Producers and Retailers

This study used MCA and Mann-Whitney U test to further understand the differences in fraud factors between two-tier groups (producers, retailers) in the baijiu supply chain. Initially, general judgments of differences were made by MCA, followed by a Mann-Whitney U test for the statistical significance of differences.

#### 4.2.1. Difference Analysis of Multiple Correspondence Analysis

The results of the differences between producers and retailers in terms of fraud vulnerability are shown in Figure 4. Figure 4a expresses the producer and retailer fraud factor vulnerability situation, where green point represents low vulnerability, blue point represents medium vulnerability, and red point represents high vulnerability. Figure 4b expresses the clustering of the production and retail sectors of the baijiu supply chain, where blue point represents producers and red point represents retailers. The first two dimensions in the MCA explain 55.063% of the total variance (Figure 4). From Figure 4a, the first quadrant fraud factors are dominated by the low vulnerability of motivations and control measures. The second quadrant fraud factors are dominated by low vulnerability and high vulnerability. The third and fourth quadrant fraud factors are dominated by medium vulnerability fraud. Figure 4b shows that retailers are mainly distributed in the first quadrant, and producers are mostly distributed in the second, third, and fourth quadrants. The distribution is concentrated in the second and third quadrants and scattered in the first and fourth quadrants.

#### 4.2.2. Mann-Whitney U Test

The significance of the difference in fraud factors between producers and retailers is shown in Table 3. Of the 40 fraud factors, 26 fraud factors differed significantly between producers and retailers, including 4 opportunity fraud factors, 12 motivation fraud factors, and 10 control measure fraud factors. Among the 26 fraud factors, retailers have a higher vulnerability than producers in 8 fraud factors (Factors 4, 9, 10, 28, 33, 35, 38, and 39), while producers have a higher vulnerability than retailers in only one fraud factor (Factor 18). This indicates that retailers have a higher risk of fraud vulnerability than producers. This result corresponds to the result in Figure 3. Contrary to the results reported by Yang et al., the vulnerability of producers in the olive oil supply chain is higher than that of retailers [7]. The reason for this difference may be that baijiu and olive oil are two different products. Specific differences in the vulnerability of the baijiu supply chain are discussed in terms of opportunities, motivations, and control measures.

##### Opportunities

Both the difficulty of final product counterfeiting and historical events (Factors 4 and 9) show medium vulnerability in the production sector and are rated as a high vulnerability in the retail sector. Producers believe that baijiu counterfeiting has certain technical difficulties, while retailers believe that the technical methods of baijiu counterfeiting are extremely easy. Producers believe that dishonest events such as adulteration and counterfeiting of baijiu occur infrequently and less information is publicly available, while retailers believe that baijiu counterfeiting events are more frequent in the market and information about the events is mostly publicly disclosed. The reason for this difference may be that the retailers’ judgment is based on the frequency of baijiu fraud events in the market, and some producers tend to choose their own relatively favorable answers because of the sensitivity of the questions.

##### Motivations

Raw material price (Factor 10) is rated as a low vulnerability by producers, while retailers identify it as a high vulnerability factor. The difference in sourcing raw materials between production and retail is responsible for the difference in the vulnerability of the fraud factor across sectors. The raw materials for baijiu are crops such as corn and sorghum. From the data obtained from producers, 32% of producers believe that raw material prices often rise, 27.1% of producers believe that raw material market prices fluctuate greatly, 39.5% of producers believe that baijiu raw material prices have remained relatively stable, and the fluctuations are not large, and only 1.2% of producers believe that baijiu raw material prices often fall. Retailers purchase products that have been packaged with finished baijiu. Up to 44.4% of them believe that the market price of baijiu often rises, and a large number of retailers sell products that are well-known domestic baijiu brands with high prices. The economic situation of the supplier (Factor 18) is identified as a medium vulnerability in the producer sector and a low vulnerability in the retail sector. The reason for this difference may be that the raw material suppliers of baijiu producers are grain producers and purchasers, and their profits are relatively small. The upstream suppliers of retailers are distributors or agents who deal with baijiu producers and then distribute them into the market. As the supply chain of baijiu continues to extend, the price of baijiu will gradually climb. In comparison, the business entities that provide raw materials for baijiu are less profitable than those in the distribution sector, so their economic situation is more fragile.

##### Control Measures

The raw material counterfeiting detection system (Factor 28) is considered to be a low vulnerability in the producer sector and a high vulnerability in the retailer sector. The reason may be that baijiu producers have more perfect counterfeiting detection processes and technologies than retailers. The production sector pays more attention to the quality control of baijiu. After testing the product quality and safety, anti-counterfeit labels are set up, and records are kept for qualified products. In the retail sector, most retailers do not test the authenticity of their products when purchasing baijiu products and do not have the appropriate testing equipment and technology, so they are more vulnerable in the detection of counterfeiting. Supplier fraud prevention management process and the local law enforcement against food fraud (Factors 35 and 39) are both low vulnerability in the production sector but medium vulnerability in the retailer sector. Suppliers of raw materials for baijiu pay more attention to product quality control and testing than enterprises in the distribution sector. Distributors and agents in the distribution sector purchase products directly from baijiu producers and lack specialized tools for testing food quality. Retailers generally respond to the infrequency and low punishment of local enforcement against food fraud. Current enforcement activities against food fraud are more stringent in the baijiu production sector, while regulatory enforcement in the retail sector needs to be strengthened. In addition, the vulnerability of the enterprise’s internal whistleblowing system and fraud prevention measures (Factors 33 and 38) in different supply chain sectors are also quite different. Most small baijiu retailers with 2–3 employees are easier to manage, so most do not have an internal whistle-blowing system for unethical behavior of internal employees; however, the production enterprise involves a number of complex processes such as baijiu brewing, processing, packaging, etc. The management process is more cumbersome, and there are greater opportunities for employees to commit fraud and other unethical acts. To prevent these acts and protect the interests of the enterprise, more producers have set up internal whistle-blowing channels. Compared with the production sector, the retail sector of baijiu lacks the corresponding guidance scheme to prevent food fraud.

## 5. Conclusions and Implications

### 5.1. Conclusions

This study assesses the vulnerability of fraud factors in the Chinese baijiu supply chain in order to understand the differences in perceived fraud vulnerability between tiers. The main findings are as follows: first, fraud factors related to opportunities in the baijiu supply chain are generally medium vulnerability, with technical opportunity vulnerability being higher than opportunities in time and space vulnerability. Second, fraud factors related to motivations in the baijiu supply chain are generally low to medium vulnerability, with economic drivers being more vulnerable than cultural and behavioral drivers. Third, fraud factors related to control measures in the baijiu supply chain are generally low to medium vulnerability. For producers, the vulnerability of control measures fraud factors is low overall, and control measures are completer and more reasonable; however, for retailers, some managerial measures need to be strengthened. Fourth, compared with production, the risk of fraud in baijiu retail is higher.

### 5.2. Theoretical Implications

First, this study uses a mature food fraud vulnerability assessment tool to investigate the fraud vulnerability of the Chinese baijiu supply chain and innovatively constructs a food fraud vulnerability assessment system for Chinese baijiu, enriching the theoretical research related to the field of food fraud in China. Second, this study uses routine activities theory from criminology in an interdisciplinary manner to analyze the main causes of the high incidence of fraud in the baijiu industry, opening up a direction for existing food safety management research. Third, this study proposes countermeasures to mitigate the risk of food fraud in China’s baijiu industry in response to the results of the food fraud vulnerability assessment, which promotes research on food fraud risk governance and enriches the research thinking on food safety risk governance in China.

### 5.3. Practice Implications

First, enterprises and related organizations should strengthen technological innovation to reduce the opportunity for fraud. The occurrence of food fraud would be reduced at the technical level by raising the technical threshold of food fraud. The organizations could also establish a food fraud knowledge base and a database of related events, which would help improve the technical level in potential food fraud risk assessment, risk identification, information communication, information traceability, and other aspects, thereby reducing the technical opportunities for illegal businesses to commit fraud. Second, enterprises and related organizations should strengthen multi-party management and weaken the motivation of food fraud. In addition to fulfilling laws and regulations and corresponding food production and operation standards, enterprises in all sectors of the food supply chain should also have a high degree of corporate social responsibility to jointly prevent and resist the occurrence of dishonest production and operation behaviors such as food fraud. Third, we should build a multi-body governance pattern to improve control measures. Food fraud governance requires the participation of multiple subjects in all aspects. Through the benign complementary advantages of all parties, the risk of food fraud would be minimized.

### 5.4. Limitations

First, the food fraud vulnerability assessment system established in this study takes opportunities, motivations, and control measures as the primary indicators and 40 secondary fraud factor indicators to measure. In the future, the indicators can be further subdivided and expanded to ensure more scientific research. Second, the suggestions on the governance of food fraud proposed in this study are mainly launched from the three elements of fraud, but the governance of food fraud involves multiple subjects and aspects. In the future, the problem of food fraud can be analyzed and discussed from other perspectives. Third, due to limited conditions, the number of retailers surveyed is relatively small. Therefore, the follow-up research could continue to supplement and improve the vulnerability assessment results of food fraud.

## Figures and Tables

**Figure 1 foods-12-00516-f001:**
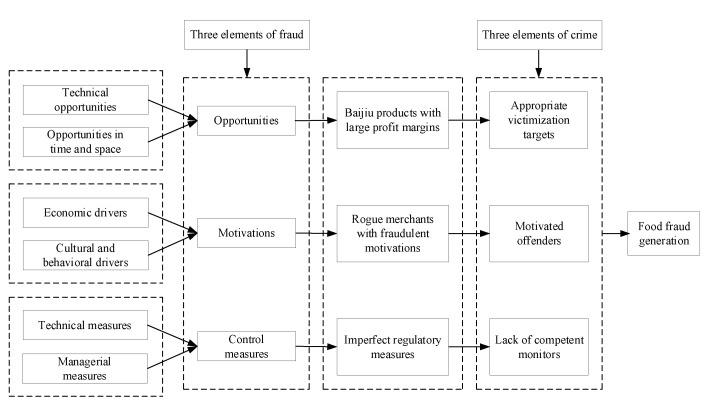
Theoretical framework chart.

**Figure 2 foods-12-00516-f002:**
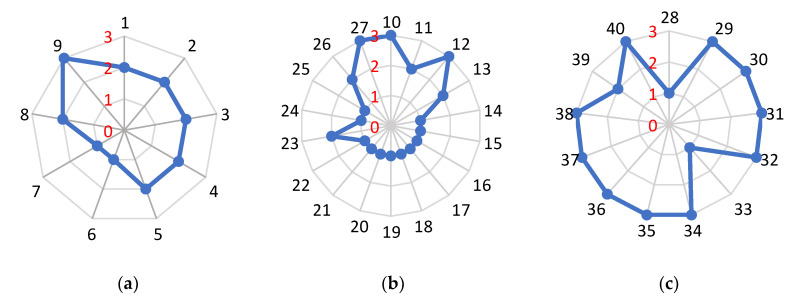
The radar charts for opportunities, motivations, and control measures. Notes: (**a**), (**b**), and (**c**) present the overall vulnerability of the opportunities, motivations, and control measures, respectively; The black numbers represent the 40 fraud factors; The red numbers represent the vulnerability level.

**Figure 3 foods-12-00516-f003:**
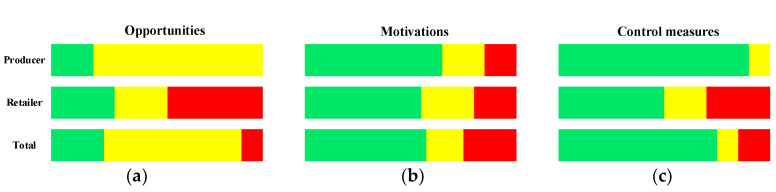
The percentage of vulnerability to different fraud factors in the baijiu supply chain. Note: (**a**), (**b**), and (**c**) present the vulnerability of the opportunities, motivations, and control measures for producer, retailer, and total, respectively; Green, yellow, and red represent low, medium, and high vulnerabilities, respectively.

**Figure 4 foods-12-00516-f004:**
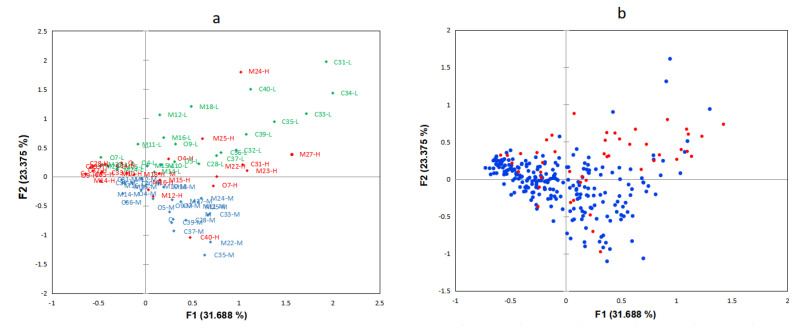
(**a**) Loading plot (**left**) and (**b**) score plot (**right**) of MCA on the food fraud vulnerability of producers and retailers. Notes: In the loading diagram, the letter O before the number represents opportunity, M represents motivation, C represents control measures, and the letters L, M, and H after the number represent low, medium, and high vulnerability, respectively.

**Table 1 foods-12-00516-t001:** The adjusted baijiu supply chain assessment index system.

Opportunities	Motivations	Control
Technical opportunities	Economic drivers	Technical measures
1. The degree of difficulty of adulteration of raw materials	10. Raw material price	28. Raw material counterfeit detection system
2. Raw material adulteration technology and knowledge accessibility	11. Raw material supply	29. Final product counterfeit detection system
3. Ability to detect adulteration of raw materials	12. The special composition or properties of raw materials	30. Traceability system for enterprises
4. The degree of difficulty in counterfeiting the final product	13. The economic situation of the enterprise	35. Fraud prevention management process for suppliers
5. Ability to detect final product counterfeiting	17. Financial pressure on suppliers	36. Supplier’s traceability system
Opportunities in time and place	18. Economic situation of suppliers	40. Fraud contingency programs
6. Control of production and processing process	23. The economic situation of the supply chain sector	Managerial measures
7. Supply chain transparency	26. Historical events of fraud in the supply chain	31. Integrity screening of employees
8. Historical events of raw material counterfeiting	27. Competition level in the supply chain	32. Ethical and cultural code of conduct for enterprises
9. Historical events of final product counterfeiting	Cultural and behavioral drivers	33. Internal whistleblowing system of the enterprise
	14. Organizational strategy of enterprises	34. Contract requirements with suppliers
	15. The ethical culture of enterprises	37. The degree of social control of the supply chain
	16. Violation of laws and regulations of enterprises	38. Fraud prevention measures in the supply chain
	19. Supplier’s organizational strategy	39. Local enforcement against food fraud
	20. Ethical culture of suppliers	
	21. Violations of laws and regulations of suppliers	
	22. Suppliers’ exposure to food fraud	
	24. Crime situation in the downstream chain	
	25. The ethical culture of fraud in the supply chain	

**Table 2 foods-12-00516-t002:** Number of interviewees statistics.

Research Links	Interviewee Position	Number	Percentage
Producer	Middle and senior managers	172	70.8%
	Employees in the production departments	37	15.2%
	Employees in other departments	34	14.0%
Retailers	Middle and senior managers	45	100.0%

**Table 3 foods-12-00516-t003:** Vulnerability of different groups to fraud factors and Mann-Whitney U test.

Key Elements	Fraud Factors	Producers	Retailers	Rank Average of Producers	Rank Average of Retailers	Z-Value	Sig. (2-Tailed)
Opportunities	Factor 1	M	NA				
	Factor 2	M	NA				
	Factor 3	M	NA				
	Factor 4	M	H	134.49	198.56	−5.252	0.000 ***
	Factor 5	M	M	147.63	127.62	−2.006	0.013 **
	Factor 6	L	NA				
	Factor 7	L	L	145.03	141.66	−2.774	0.005 ***
	Factor 8	M	NA				
	Factor 9	M	H	159.97	60.94	−7.820	0.000 ***
Motivations	Factor 10	L	H	146.44	134.00	−2.001	0.027 **
	Factor 11	M	M	146.51	133.66	−3.206	0.000 ***
	Factor 12	H	NA				
	Factor 13	M	M	150.84	110.28	−4.517	0.000 ***
	Factor 14	L	NA				
	Factor 15	L	NA				
	Factor 16	L	NA				
	Factor 17	L	L	143.60	149.37	−3.568	0.000 ***
	Factor 18	M	L	146.90	131.53	−2.229	0.009 **
	Factor 19	L	L	142.55	155.03	−2.007	0.022 **
	Factor 20	L	L	139.35	172.32	−2.862	0.004 ***
	Factor 21	L	L	141.06	163.06	−1.990	0.047 **
	Factor 22	L	L	143.14	151.83	−2.775	0.004 ***
	Factor 23	M	M	152.36	102.04	−4.126	0.000 ***
	Factor 24	L	NA				
	Factor 25	L	L	139.60	170.93	−2.697	0.007 ***
	Factor 26	NA	M				
	Factor 27	H	H	142.66	154.41	−2.029	0.019 **
Control measures	Factor 28	L	H	150.14	114.07	−2.891	0.004 ***
	Factor 29	L	NA				
	Factor 30	L	NA				
	Factor 31	L	L	149.27	118.72	−2.639	0.008 ***
	Factor 32	L	L	152.54	101.09	−4.644	0.000 ***
	Factor 33	M	H	149.69	116.48	−2.608	0.009 ***
	Factor 34	L	L	145.91	136.91	−2.725	0.006 ***
	Factor 35	L	M	144.82	142.78	−4.231	0.000 ***
	Factor 36	L	L	142.78	142.70	−3.233	0.000 ***
	Factor 37	L	NA				
	Factor 38	L	H	154.93	88.20	−5.449	0.000 ***
	Factor 39	L	M	143.19	151.60	−1.993	0.049 **
	Factor 40	L	L	152.13	103.28	−4.382	0.000 ***

Notes: Low, medium, and high vulnerability are coded with L, M, and H, respectively; NA indicates that the indicator is not applicable to the interviewees; ** and *** indicate significance at the 5% and 1% significance levels, respectively.

## Data Availability

The data presented in this study are available on request from the corresponding author.

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
