# Peer review of "Food Fraud Vulnerability Assessment in the Chinese Baijiu Supply Chain"

_foods, 2023, doi:10.3390/foods12030516_

Round 1
Reviewer 1 Report
The article is fascinating and good scientific work, but it needs some improvement; these are:
1. The Article needs to do a literature review to find a research gap to justify the novelty of the research and develop a theoretical framework.
2. The discussion section needs comparisons with relevant studies.
3. The end of the paper needs to elaborate on the research implication to theoretical development and managerial/policy practice.
Reviewer 2 Report
The manuscript by Wang et al. is well written and within the aims and scope of foods and is well written with little to no grammatical mistakes. The article is about Chinese baijiu, and how to prevent fraud vulnerability. The manuscript uses the food fraud vulnerability assessment and the safe supply of affordable food everywhere to evaluate the vulnerability of fraud in baijiu. The article is well written
The following article should be included in this manuscript as it pertains directly to this manuscript :
Burns, Rachel "A Fast, Straightforward and Inexpensive Method for the Authentication of Baijiu Spirit Samples by Fluorescence Spectroscopy." Beverages 7, no. 3 (2021): 65.
The aforementioned article goes over an analytical technique for the detection of fraud in baijiu samples. This article should be cited in this manuscript. This article even goes in detail how inexpensive baijiu can be mixed with expensive baijiu, however fluorescence spectroscopy can detect this type of adulteration. This citation should go in section 4.1.1.1 under technical opportunities.
Lines 232-233. As the authors state “baijiu counterfeiting is easier while detection is more difficult.” I would like to challenge this statement with the above reference.
I would be willing to accept this manuscript if the authors can address my questions and concerns regarding there statement that detection of counterfeit baijiu is difficult. I would argue that is easy to detect counterfeit baijiu liquor samples.
Round 2
Reviewer 2 Report
I would like to thank the authors for making the necessary corrections to this manuscript. The authors have satisfied my requests and I approve this manuscript for publication.